# Label-efficient Training of Small Task-specific Models by Leveraging Vision Foundation Models

## Abstract

Large Vision Foundation Models (VFMs) pretrained on massive datasets exhibit impressive perform on various downstream tasks, especially with limited labeled target data. However, due to their high memory and compute requirements, these models cannot be deployed in resource constrained settings. This raises an important question: *How can we utilize the knowledge from a large VFM to train a small task-specific model for a new target task with limited labeled training data?* In this work, we answer this question by proposing a simple yet highly effective task-oriented knowledge transfer approach to leverage pretrained VFMs for effective training of small task-specific models. Our experimental results on three target tasks under limited labeled data settings show that the proposed knowledge transfer approach outperforms task-agnostic VFM distillation, web-scale CLIP pretraining and supervised ImageNet pretraining approaches by 1-10.5%, 2-21%, and 2-14%, respectively. We also show that the dataset used for transferring knowledge has a significant effect on the final target task performance, and propose a retrieval-based approach to curate effective transfer sets.

## 1 Introduction

Currently, the computer vision community is witnessing the emergence of various vision and multi-modal foundation models pretrained on massive datasets (Radford et al., 2021; Yuan et al., 2021; Alayrac et al., 2022; Kirillov et al., 2023; Oquab et al., 2023; Li et al., 2023b; Wang et al., 2023b). These models have been shown to work well for many downstream computer vision tasks, especially, when task-specific labeled data is limited (Radford et al., 2021). While a single large foundation model could serve many applications, it cannot be directly used in resource constrained settings due to its high memory and compute requirements. Also, many real-world applications such as autonomous driving, medical image diagnostics, and industrial automation, focus on specific tasks and need small task-specific models rather than a large foundation model. This raises an important question: *How can we utilize the knowledge from a large Vision Foundation Model (VFM) to effectively train a small task-specific model for a new target task with limited labeled training data?*

Answering this question requires transferring knowledge from a VFM across both task and model architecture boundaries. This is different from knowledge distillation setting that only focuses on knowledge transfer between model architectures (Hinton et al., 2015; Tian et al., 2020) and the transfer learning setting that only focuses on knowledge transfer between tasks (Lu et al., 2021).

## 2 Approach and contributions

In this work, we propose a simple yet highly effective approach for training a small task-specific model by transferring knowledge from a large VFM. This approach, referred to as **task-oriented knowledge transfer**, first teaches the target task to the VFM by finetuning it with an appropriate task-specific head using labeled target task data, and then transfers task-oriented knowledge from the finetuned VFM to the target model using the knowledge distillation framework of Hinton et al. (2015) with a large unlabeled dataset, referred to as the *transfer set*. Finally, the target model is finetuned with labeled target task data (see Fig. 1 top).

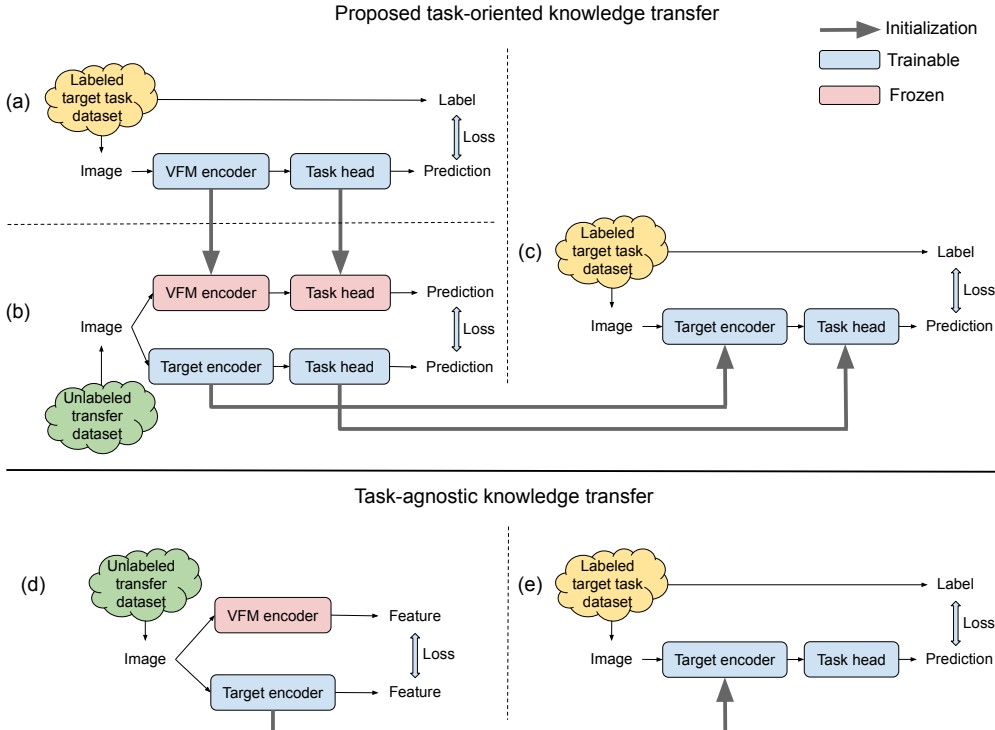

Figure 1: **Top:** Proposed task-oriented knowledge transfer approach that (a) first finetunes a VFM using labeled target task data, (b) then uses this finetuned VFM to pretrain the target model by matching their task predictions on an unlabeled transfer dataset, and (c) finally finetunes the target model using labeled target task data. **Bottom:** Alternative task-agnostic knowledge transfer approach that (d) first pretrains the target model by matching its features to the features extracted by a VFM on an unlabeled transfer dataset, and (e) then finetunes it using labeled target task data.

An alternative approach to train a small task-specific model by leveraging a VFM is to first distill the frozen VFM image encoder to the target model image encoder and then finetune the target model with appropriate task-specific head using labeled target task data (see Fig. 1 bottom). We refer to this approach as ***task-agnostic knowledge transfer***. Both task-oriented and task-agnostic knowledge transfer approaches leverage VFMs that have been trained on web-scale datasets. Instead, one could pretrain the small task-specific model directly on a web-scale dataset using the CLIP approach (Radford et al., 2021). However, direct pretraining on web-scale datasets is extremely expensive. For example, training the MobileViT-V2 model on 1.1B image-text pairs for 15 epochs took us 4.5 days with 256 A100 GPUs.

We compare the proposed task-oriented knowledge transfer from VFMs with task-agnostic transfer from VFMs, direct web-scale CLIP pretraining of the target model with 1.1B image-text pairs, and the widely-used supervised ImageNet-1k (Deng et al., 2009) pretraining of the target model on three target tasks, namely Places365 scene classification (Zhou et al., 2014), HAM10K skin lesion classification (Tschandl, 2018), and ADE20K semantic segmentation (Zhou et al., 2017) under limited labeled data settings. Specifically, we experiment with two VFMs, namely DINOV2-ViT-L/14 (Oquab et al., 2023) and OpenCLIP-ViT-L/14 (Ilharco et al., 2021), and two target mobile architectures, namely MobileViT-V2 (Mehta & Rastegari, 2023) and FastViT-S12 (Vasu et al., 2023), and present the following insightful conclusions to the community:

- Task-oriented knowledge transfer from VFMs outperforms task-agnostic transfer by a significant margin (1-10.5%). While VFMs can store vast knowledge by virtue of their large capacity, small models may not be able to inherit this vast knowledge due to their limited capacity. Hence, transferring only task-oriented knowledge is more effective.

- Both task-oriented and task-agnostic knowledge transfer from VFMs outperform the significantly more compute-intensive web-scale CLIP pretraining with 1.1B image-text pairs. Using task-oriented transfer, we observe significant performance gains in the range of 2-21%. We conjecture that this is because VFMs compress the knowledge in a large-scale dataset such that it is easy for small models to acquire this knowledge by mimicking VFMs when compared to learning directly from the original large-scale dataset.

- Both task-oriented and task-agnostic knowledge transfer from VFMs outperform the popular supervised ImageNet-1k pretraining. Using task-oriented transfer, we observe significant performance gains in the range of 2-14%.

- Transfer set has a significant effect on the final target task performance. Using a transfer set whose image distribution is close to the target task image distribution performs significantly better than using a generic image dataset such as CC3M (Sharma et al., 2018). For target-oriented transfer, we observe performance gains in the range of 0.6-5% when using task-related transfer sets. Due to their limited capacity, small models may not be able to mimic VFMs on the entire input image space. Hence, focusing on the subset of the input space that is relevant for the target task is more effective.

While using a large target task-related transfer set is better, such a dataset may not be readily available for some target tasks. In this work, we propose to address this issue by curating task-related transfer sets using image retrieval. Specifically, we use the images from the limited labeled target task dataset as queries and retrieve similar images from a large pool of images sourced from the web. Our experimental results on ADE20K segmentation dataset show that using these curated transfer sets for task-oriented knowledge transfer improves the segmentation performance by 0.6-1.8% when compared to using the generic CC3M transfer set.

## 3 EXPERIMENTAL ANALYSIS

### 3.1 EXPERIMENTAL SETUP

Our goal is to train a small target model for a specific target task. We assume that we have access to a small labeled target task dataset, a relatively large unlabeled dataset, and a pretrained VFM. We perform the target task model training in two stages: *pretraining* followed by *finetuning*. In the pretraining stage, we utilize the VFM by following the task-oriented and task-agnostic knowledge transfer approaches presented in Sec. 2 using the large unlabeled dataset as the transfer set. In the finetuning stage, we train the model on the small labeled target task dataset.

**Alternative approaches:** *IM-Pretrain*: We pretrain the target image encoder on 1.28M labeled training images from the ImageNet-1K dataset (Deng et al., 2009) using the standard cross-entropy loss. *CLIP-Pretrain*: We pretrain the target image encoder on an internal dataset of 1.1B image-text pairs using contrastive loss similar to CLIP (Radford et al., 2021).

**Target task datasets:** We use three target task datasets for evaluation. To study the effectiveness under limited labeled data settings, for each task, we conduct experiments by limiting the amount of labeled target task data used for training.

- **Places365 scene classification** (Zhou et al., 2014): This dataset has 1.8M training and 36.5K validation images. We split the original validation set into two subsets consisting of 3.65K and 32.85K images, and use them for validation and testing, respectively.

- **HAM10K skin lesion disease classification** (Tschandl, 2018): This dataset consists of 10K training, 193 validation and 1.5K test images.

- **ADE20K semantic segmentation** (Zhou et al., 2017): This dataset consists of 20.2K training and 2K validation images. We split the original training set into two subsets with 19.2K and 1K images, and use them for training and validation, respectively. We use the original 2K validation set as the test set.

**Evaluation metrics:** We use top-1 accuracy for Places365 and HAM10K classification tasks, and mean Intersection over Union (IoU) for the ADE20K segmentation task.

**Transfer sets**: For each target task, we experiment with two transfer sets. The first one is a generic transfer set consisting of 2.87M unlabeled images from the training split of the CC3M dataset (Sharma et al., 2018), and the second one is a task-related transfer set consisting of unlabeled images from the target task domain. For each task, we use the entire training split of the corresponding dataset as the unlabeled task-related transfer set, which contains 1.8M images for Places365 classification, 19.2K images for ADE20K segmentation, and 10K images for HAM10K classification.

**Foundation models:** We use the DINOV2-ViT-L/14 model (Oquab et al., 2023) and the OpenCLIP-ViT-L/14 model (Ilharco et al., 2021) trained on the DataComp-1B dataset (Samir Yitzhak Gadre, 2023) as VFMs. For brevity, we refer to them as DINOV2 and OpenCLIP, respectively.

**Target models:** We use two recent efficient architectures, namely MobileViT-V2-1.0 (Mehta & Rastegari, 2023) and FastViT-S12 (Vasu et al., 2023), as image encoders for the target models.

**Task-specific heads:** We use a linear classifier as the task-specific head for classification tasks, and a DeepLabV3 head (Chen et al., 2017) as the task-specific head for segmentation tasks. Please refer to Appendix A.1 for further details.

**Loss functions:** For finetuning with labeled target task dataset, we use the standard cross entropy loss, and for matching task predictions, we use KL divergence between the softmax outputs of VFM and target model. For segmentation tasks, these losses are used at each pixel. The loss function used for matching features depends on the VFM. In the case of OpenCLIP model, we use contrastive loss (Tian et al., 2020) with a linear projection layer on top of the target model output to match its dimensionality with the CLIP embedding dimensionality. Since DINOV2 is trained to produce good patch features along with global image features, we experiment with both image-level and patch-level features in the case of DINOV2. When using global image features for knowledge transfer, we use contrastive loss with linear projection layers on outputs of both models. When using patch features for knowledge transfer, we use cosine similarity loss with a linear projection layer on top of the target model features for dimensionality matching. We also resize DINOV2 patch features to match the spatial resolution of the target model features.

**Training details:** We use the AdamW optimizer (Loshchilov & Hutter, 2019) with cosine learning rate decay in all our experiments. Following Mehta & Rastegari (2023), we use various advanced image augmentations in our training. We use input resolutions of $256 \times 256$ and $512 \times 512$ for classification and segmentation tasks, respectively. Please see Appendix A.2 for additional training details such as batch size, learning rate and number of epochs.

We run each finetuning experiment three times and report the average results.

## 3.2 PLACES365 SCENE CLASSIFICATION

Figure 2(a) compares various approaches in terms of the final classification accuracy for different combinations of VFM, target model and transfer set. For task-agnostic transfer from DINOV2, global image features worked better than patch features. Hence, we report results corresponding to global image features in Fig. 2. Please see Tab. 2 in the Appendix for results with patch features.

Both task-oriented and task-agnostic knowledge transfer from VFMs outperform ImageNet and CLIP pretraining approaches, and task-oriented transfer performs the best. When using the generic CC3M transfer set, task-oriented transfer from OpenCLIP outperforms ImageNet pretraining by 2-8% margin, CLIP pretraining by 2-5.5% margin, and the corresponding task-agnostic transfer by 2-4.5% margin. When using the task-related Places365 transfer set, task-oriented transfer from OpenCLIP outperforms ImageNet pretraining by 3.5-10% margin, CLIP pretraining by 3.5-7.5% margin, and the corresponding task-agnostic transfer by 1.5-3% margin.

When task-related Places365 transfer set is used, task-oriented transfer even outperforms the corresponding finetuned VFM when the amount of labeled data is small. This is because the knowledge transfer process leverages unlabeled data that is not used while finetuning the VFM.

Figure 2(b) shows the improvement in accuracy when task-related Places365 transfer set is used instead of generic CC3M transfer set for task-oriented knowledge transfer. Using task-related transfer set improves the accuracy significantly (1-2.3%). We also observe such performance improvements for task-agnostic knowledge transfer (see Tab. 2 in the Appendix).

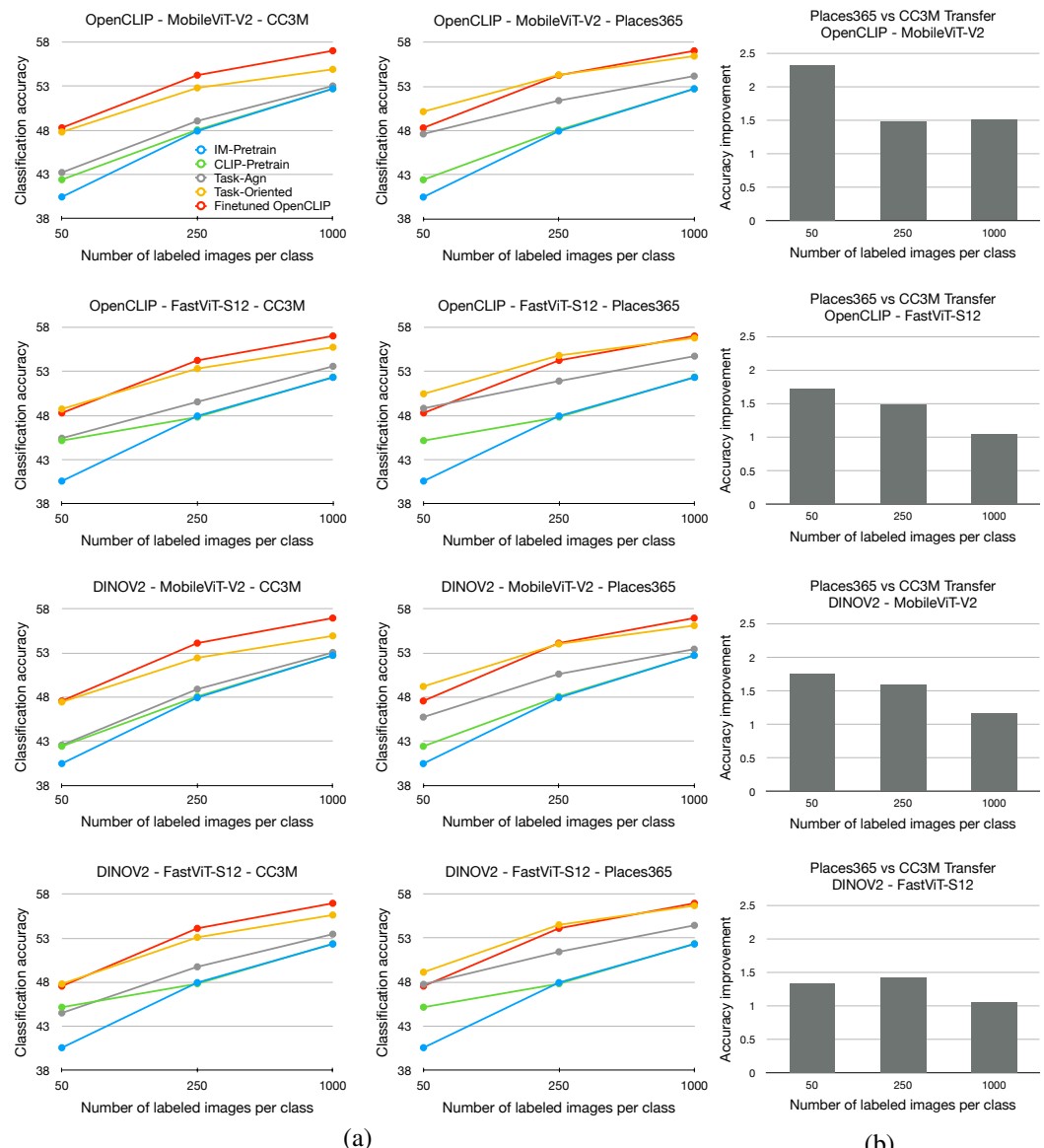

Figure 2: (a) Places365 accuracy for different VFM - Target model - Transfer set combinations. Task-oriented transfer (yellow) outperforms task-agnostic transfer, ImageNet pretraining and CLIP pretraining by a significant margin. (b) Performance improvement when Places365 is used instead of CC3M as transfer set for task-oriented knowledge transfer.

### 3.3 HAM10K Skin Lesion Disease Classification

HAM10K dataset is highly imbalanced with just 115 training images in the smallest class and 6705 training images in the largest class. When experimenting with $N$ training images per class, if a class does not have $N$ images, we just use all the images from that class.

Figure 3(a) compares various approaches for different combinations of VFM and transfer set with FastViT-S12 as the target model. For task-agnostic knowledge transfer from DINOV2, we experimented with both global image features and patch features and present the best results in Fig. 3. Please see Tab. 1 in the Appendix for full results.

Task-oriented transfer performs the best, and task-agnostic transfer outperforms ImageNet and CLIP pretraining approaches. When using the generic CC3M transfer set, task-oriented transfer from

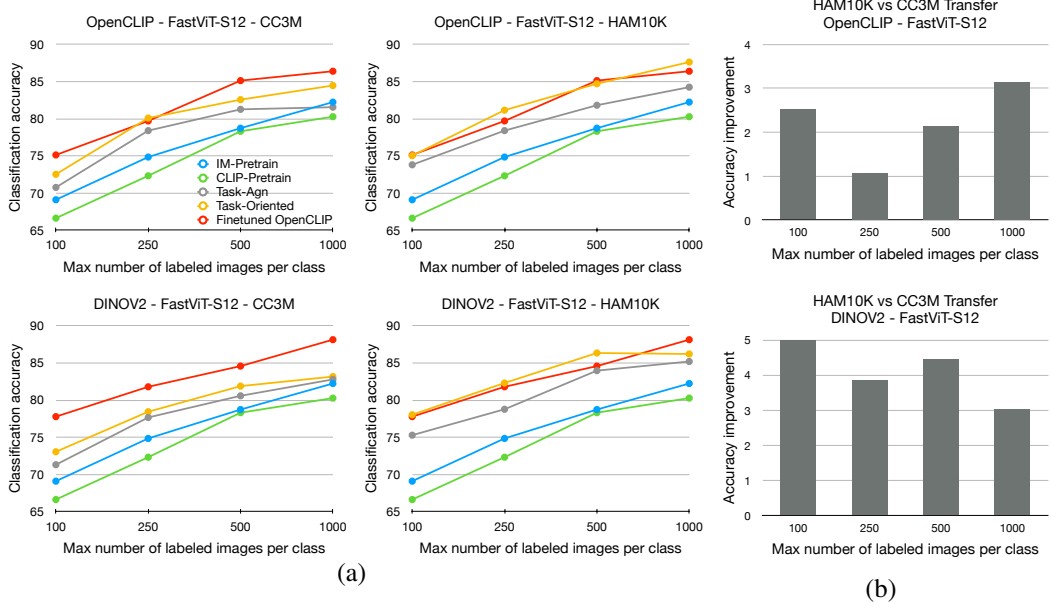

Figure 3: (a) HAM10K classification results for different combinations of VFM and transfer set with FastViT-S12 as the target model. Task-oriented transfer (yellow) outperforms task-agnostic transfer, ImageNet pretraining and CLIP pretraining by a significant margin. (b) Accuracy improvement when HAM10K is used instead of CC3M as transfer set for task-oriented knowledge transfer.

OpenCLIP outperforms ImageNet pretraining by 2-5% margin, CLIP pretraining by 4-7.5% margin, and the corresponding task-agnostic transfer by 1.3-3% margin. When using the task-related HAM10K transfer set, task-oriented knowledge transfer from DINOV2 outperforms ImageNet pretraining by 4-9% margin, CLIP pretraining by 6-11.5% margin, and the corresponding task-agnostic transfer by 1-3.5% margin. Similar to the Places365 results, task-oriented transfer with HAM10K transfer set outperforms the corresponding finetuned VFM in some cases.

Figure 3(b) shows performance improvements for task-oriented transfer when task-related HAM10K transfer set is used instead of generic CC3M transfer set. Using task-related transfer set leads to large improvements (1-5%) . It is worth highlighting that HAM10K transfer set contains only 10K images and still outperforms CC3M transfer set that has 2.87M images. This underscores the importance of the relevance of transfer set to the target task.

## 3.4 ADE20K SEMANTIC SEGMENTATION

Figure 4(a) presents the final ADE20K semantic segmentation results for different combinations of target model and transfer set with DINOV2 as the VFM. As DINOV2 is explicitly trained to produce good patch features, we observed that knowledge transfer from DINOV2 performs significantly better than transfer from OpenCLIP in our preliminary experiments. Hence, we only present knowledge transfer results corresponding to DINOV2 for this task. We also observed that, for task-agnostic transfer from DINOV2, using patch features performs significantly better than global image features. Hence, we only report task-agnostic transfer results with patch features. Please see Tab. 3 in the Appendix for the mean IOU values corresponding to Fig. 4.

Similar to the Places365 and HAM10K classification results, task-oriented knowledge transfer performs the best, and task-agnostic transfer outperforms ImageNet and CLIP pretraining approaches. When using the generic CC3M transfer set, task-oriented transfer outperforms ImageNet pretraining by 5-14% margin, CLIP pretraining by 6.5-21% margin, and the corresponding task-agnostic transfer by 3.5-10.5% margin. When using the task-related ADE20K transfer set, task-oriented transfer outperforms ImageNet pretraining by 3.5-12% margin, CLIP pretraining by 5.5-19% margin, and the corresponding task-agnostic transfer by 1-6% margin.

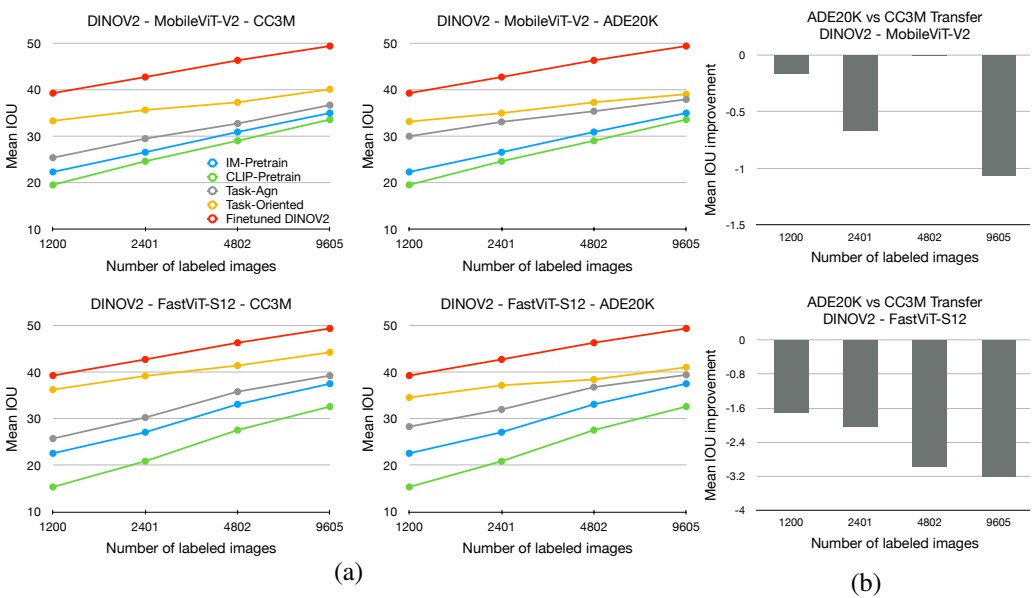

Figure 4: (a) ADE20K segmentation results for different combinations of target model and transfer set with DINOV2 as the VFM. Task-oriented transfer (yellow) outperforms task-agnostic transfer, ImageNet pretraining and CLIP pretraining by a significant margin. (b) Performance improvement when ADE20K is used instead of CC3M as transfer set for task-oriented knowledge transfer.

Figure 4(b) shows performance improvements for task-oriented transfer when task-related ADE20K transfer set is used instead of generic CC3M transfer set. While using task-related transfer sets improves the performance in the case of Places365 and HAM10K classification tasks, it performs worse than generic CC3M transfer set in the case of ADE20K segmentation. We conjecture that the main reason for this is the size of ADE20K transfer set which has only 19K images. We address this issue by curating a large task-related transfer set using image retrieval as shown in the next section.

## 3.5 TRANSFER SET CURATION

Our results on Places365 and HAM10K datasets show that using task-related transfer set performs better than a generic transfer set such as CC3M if the task-related transfer set is sufficiently large [1]. However, such large transfer sets may not be readily available for some tasks. In this section, we curate task-related transfer sets using image retrieval and demonstrate their effectiveness for the task of ADE20K segmentation. Specifically, we use the limited target task dataset as the query set $\mathcal{Q}$ and YFCC15M dataset Radford et al. (2021) which contains 15M images filtered from the original YFCC100M dataset (Thomee et al., 2016) as the gallery $\mathcal{G}$. We use OpenCLIP-ViT-L/14 image encoder (Ilharco et al., 2021) trained on the DataComp-1B dataset (Samir Yitzhak Gadre, 2023) as an encoder network $\phi$ to map all the images to a $d$-dimensional embedding space, and perform retrieval based on Euclidean distances in this space. We explore the following retrieval strategies:

- **Random:** Randomly select images from the gallery.
- **Best-matches**: For each image $x \in \mathcal{G}$, we use $\min_{x_q \in \mathcal{Q}} \|\phi(x) - \phi(x_q)\|_2$ as its distance to the query set $\mathcal{Q}$. We retrieve images from $\mathcal{G}$ in the increasing order of their distance to $\mathcal{Q}$.
- **Query-balanced (Image)**: For a query image $x_q \in \mathcal{Q}$, we define $k$-NN$(x_q)$ to be the set of its $k$ nearest neighbors from the gallery $\mathcal{G}$. To retrieve $N$ images in total, we find the smallest $k$ for which $\bigcup_{x_q \in \mathcal{Q}} k$-NN$(x_q)$ contains at least $N$ images. If $\bigcup_{x_q \in \mathcal{Q}} k$-NN$(x_q)$ contains more than $N$ images, we drop the $k^{th}$ neighbor of randomly selected queries until the retrieved set contains $N$ images.

---

[1]The size of task-related transfer set needed to outperform CC3M transfer depends on the task. While 10K images are sufficient for HAM10K classification, even 19K images are insufficient for ADE20K segmentation.

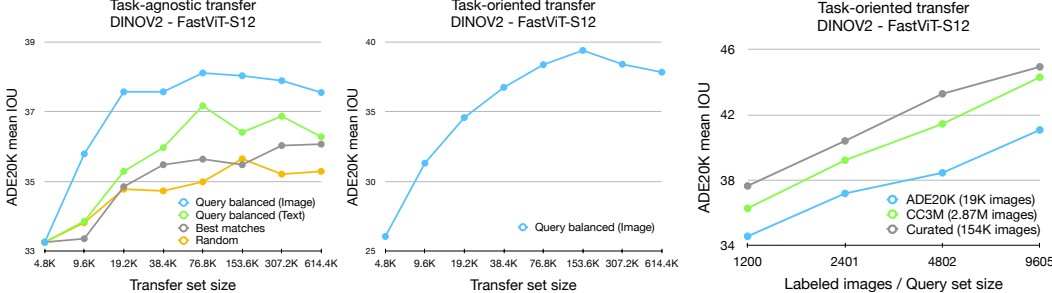

Figure 5: **Left:** Performance of task-agnostic transfer using transfer sets curated by different retrieval strategies. **Middle:** Performance of task-oriented transfer using transfer sets curated using the image query-balanced retrieval strategy. Here, we use 4800 labeled ADE20K images both as the finetuning dataset and the query set. **Right:** Performance of task-oriented transfer for various transfer sets. Curated transfer sets outperform both CC3M and ADE20K transfer sets.

- **Query-balanced (Text):** First, we convert the class names in ADE20K dataset into text descriptions using the templates from Radford et al. (2021) and encode these text descriptions using the text encoder of the OpenCLIP model used to encode images. Then, we follow the above query balanced retrieval strategy using text queries instead of image queries.

We use the combination of DINOV2 and FastViT-S12 for these experiments. Figure 5 (left) shows ADE20K segmentation performance for task-agnostic transfer using transfer sets curated by different retrieval strategies. Here, we use 4800 labeled images for finetuning the target model and use the same 4800 images as the query set for retrieval. Query-balanced retrieval based on image queries performs the best. By giving equal weight to all queries, this approach increases diversity in the retrieved samples when compared to the best-matches strategy. The segmentation performance increases with the transfer set size until we reach 77K-154K images and drops slowly after that. We performed a similar experiment for task-oriented transfer using the best performing image query-balanced retrieval strategy. Figure 5 (middle) shows the corresponding results. Again, the performance increases until we reach 154K images and drops after that indicating that the size of YFCC15M subset that is most useful for ADE20K segmentation is around 154K.

Using 154K as the target transfer set size, we curated different transfer sets by varying the number of query images used for retrieval. Figure 5 (right) compares the performance of these curated transfer sets with the CC3M and ADE20K transfer sets for task-oriented transfer. Curated transfer sets clearly outperform both task-related ADE20K and generic CC3M transfer sets. [2]

It is worth noting that by using just 4802 labeled images, we are able to achieve a mean IOU of 43.28 with FastViT-S12 backbone by leveraging pretrained VFMs and image retrieval. In contrast, a 3x larger ResNet-50 (He et al., 2016) backbone achieves a lower mean IOU of 42.42 when trained with 20K labeled images, which is 4x more labeled data (MMSegmentation, 2020).

# 4 RELATED WORKS

**Knowledge distillation** is a widely-used approach for transferring knowledge between model architectures by training one model to mimic the outputs of another model. Numerous knowledge distillation approaches have been proposed over the past decade based on various knowledge representations such as task logits (Hinton et al., 2015), intermediate features or embeddings (Heo et al., 2019; Tian et al., 2020), relations between samples (Park et al., 2019), attention maps (Zagoruyko & Komodakis, 2017), etc. Please refer to Wang & Yoon (2022); Hu et al. (2023) for an overview of existing knowledge distillation approaches. Some recent distillation works have specifically focused

---

[2]The best performance in the middle figure is lower than the performance for curated transfer set corresponding to 4800 query images in the right figure. This is because, we used shorter training runs (60K steps) to get the results in the middle figure, and once we identified the best transfer set size, we used longer training runs (200K steps) for the right figure.

on multi-modal distillation of image-language models (Fang et al., 2021; Li et al., 2023c; Wang et al., 2023a; Sun et al., 2023; Yang et al., 2023). In addition to transferring knowledge between model architectures, this work also focuses on transferring knowledge between tasks.

**Transfer learning**, where a model is first pretrained on a data-rich task before being partially or fully finetuned on a downstream task, has been well studied over the past decade (Niu et al., 2020; Lu et al., 2021), and is widely used to demonstrate the effectiveness of VFMs for several downstream tasks (Radford et al., 2021; Jia et al., 2022; Oquab et al., 2023). Recently, Entezari et al. (2023) compared various pretraining approaches and showed that supervised ImageNet training and large-scale image-text contrastive training are effective pretraining strategies for several downstream vision tasks. While the standard transfer learning setting focuses on transferring knowledge only between tasks, this work focuses on transferring knowledge between both tasks and model architectures.

**Image retrieval strategy** has been used by various recent works to curate training datasets (Udandarao et al., 2022; Li et al., 2023a; Xu et al., 2023; Wallingford et al., 2023; Liu et al., 2023). While Li et al. (2023a) focuses on self-supervised learning, the remaining works focus on training or adapting vision-language models. Different from these works, we use retrieval to curate task-related datasets used for transferring knowledge from VFMs to small task-specific models.

**Task-oriented knowledge transfer from foundation models** has been recently explored in the context of Large Language Models (LLMs) by Hsieh et al. (2023); Fu et al. (2023). These approaches use chain-of-thought prompting to extract rationales from LLMs and use these rationales to train small task-specific models. In this work, we focus on vision foundation models.

**Self-supervised learning**, which uses unlabeled data to obtain a good initial feature representation, has received significant attention in the recent past, and several approaches have been proposed based on contrastive learning (Chen et al., 2020; He et al., 2020), distillation (Grill et al., 2020; Chen & He, 2021; Caron et al., 2021), redundancy reduction (Zbontar et al., 2021), clustering (Caron et al., 2018; 2020) and image inpainting (He et al., 2022; Bao et al., 2022). Please refer to Ozbulak et al. (2023) for a detailed review of existing self-supervised learning approaches.

**Semi-supervised learning** approaches leverage both labeled and unlabeled data to improve the final task performance. They focus on effectively propagating label information from a labeled dataset to an unlabeled dataset (Lee, 2013; Xie et al., 2020b), and training the network using consistency constraints on the unlabeled samples (Laine & Aila, 2017; Tarvainen & Valpola, 2017; Berthelot et al., 2019; Xie et al., 2020a; Sohn et al., 2020; Verma et al., 2022). Please refer to Chen et al. (2022) for a recent survey of various semi-supervised approaches.

Knowledge transfer from VFMs is complimentary to self/semi-supervised learning approaches and can potentially be combined with them to further improve task-specific models.

## 5 CONCLUSIONS

In this work, we proposed a simple yet highly effective task-oriented knowledge transfer approach for training small task-specific models by leveraging pretrained VFMs. We experimented with two VFMs and two mobile target architectures, and showed that the proposed approach outperforms task-agnostic VFM distillation, web-scale CLIP pretraining and supervised ImageNet pretraining approaches by a significant margin on three target tasks. We highlighted the importance of transfer set distribution and showed that large generic transfer sets such as CC3M perform worse than much smaller task-related transfer sets. We also proposed a retrieval-based strategy to curate task-related transfer sets, and experimentally demonstrated the effectiveness of these curated transfer sets.

In this work, we only used labeled target task data to finetune VFMs. We could potentially leverage additional unlabeled data to better adapt VFMs to the target task/domain, thereby eventually improving the small task-specific model trained with knowledge transfer from VFMs. We plan to explore this in the near future.

**Limitations:** Since the proposed approach transfers task-oriented knowledge from VFMs, the target models may inherit the biases of the foundation models. Knowledge transfer from VFMs is most effective when a large task-related transfer set is available. Curating such transfer sets could be difficult for some specialized domains such as health care and industrial automation that are not well covered by web data.

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

# A EXPERIMENTAL SETUP

## A.1 TASK-SPECIFIC HEADS

For classification tasks, the task-specific head is a linear classifier. The input to the classification layer is the final CLS token features for DINOV2 and OpenCLIP models, and the output of the final average pooling layer for MobileViT-V2 and FastViT-S12 models. For segmentation tasks, the task-specific head consists of a DeepLabV3 segmentation head (Chen et al., 2017) on top of the final spatial feature map and a spatial upsampling layer. As segmentation tasks requires high resolution features, we modify the image encoders such that their output spatial resolution is $1/8$ of the input image resolution. For DINOV2 and OpenCLIP models, we use a convolution stride of 8 instead of 14 for the patch embedding layer and resize the position embeddings accordingly. For MobileViT-V2 and FastViT-S12 models, we change the last two stride-2 convolutions to stride-1 convolutions.

## A.2 TRAINING DETAILS

**Augmentations:** For downstream classification training, task-agnostic VFM distillation and classification task logits distillation, we adopt the advanced image augmentations used for ImageNet-1k in Mehta & Rastegari (2023) and an input resolution of $256 \times 256$. For downstream segmentation training and segmentation task logits distillation, we adopt the segmentation task-related augmentations used in Mehta & Rastegari (2023) and an input resolution of $512 \times 512$.

For CLIP pretraining of MobileViT-V2 and FastViT-S12 models on 1.1B image-text pairs, we use a learning rate of $5e^{-3}$ and train with a batch size of 65.5K for 200K steps. For ImageNet pretraining, we use a learning rate of $2e^{-3}$ and train with a batch size of 1024 for 300 epochs.

| Maximum labeled training images per class | | 100 | 250 | 500 | 1000 |
|---|---|---|---|---|---|
| Finetuned DINOV2 | | $77.78 \pm 0.61$ | $81.79 \pm 0.97$ | $84.57 \pm 1.88$ | $88.12 \pm 0.56$ |
| Finetuned OpenCLIP | | $75.13 \pm 0.82$ | $79.70 \pm 1.19$ | $85.12 \pm 0.35$ | $86.38 \pm 0.74$ |
| Target model - FastViT-S12 | | | | | |
| IM-Pretrain | | $69.11 \pm 1.56$ | $74.85 \pm 0.82$ | $78.73 \pm 1.88$ | $82.23 \pm 1.37$ |
| CLIP-Pretrain | | $66.64 \pm 1.75$ | $72.33 \pm 1.11$ | $78.31 \pm 1.99$ | $80.27 \pm 2.08$ |
| DINOV2 (CC3M transfer) | Task-Agn (Patch) | $71.76 \pm 2.28$ | $75.68 \pm 1.83$ | $80.34 \pm 0.74$ | $80.95 \pm 1.16$ |
| | Task-Agn (Image) | $71.32 \pm 1.94$ | $77.67 \pm 1.48$ | $80.58 \pm 0.53$ | $82.78 \pm 1.01$ |
| | Task-Oriented | $73.06 \pm 0.79$ | $78.44 \pm 0.41$ | $81.88 \pm 0.64$ | $83.16 \pm 0.83$ |
| DINOV2 (HAM10K transfer) | Task-Agn (Patch) | $75.29 \pm 1.22$ | $78.77 \pm 1.55$ | $83.96 \pm 0.10$ | $85.19 \pm 0.40$ |
| | Task-Agn (Image) | $72.18 \pm 2.57$ | $76.85 \pm 0.74$ | $81.68 \pm 0.85$ | $83.64 \pm 0.59$ |
| | Task-Oriented | $78.04 \pm 0.05$ | $82.30 \pm 0.14$ | $86.33 \pm 0.16$ | $86.20 \pm 0.32$ |
| OpenCLIP (CC3M transfer) | Task-Agn | $70.77 \pm 1.00$ | $78.40 \pm 0.77$ | $81.26 \pm 0.37$ | $81.55 \pm 1.31$ |
| | Task-Oriented | $72.53 \pm 0.63$ | $80.09 \pm 0.24$ | $82.56 \pm 0.4$ | $84.46 \pm 0.65$ |
| OpenCLIP (HAM10K transfer) | Task-Agn | $73.81 \pm 0.44$ | $78.40 \pm 0.27$ | $81.81 \pm 0.82$ | $84.24 \pm 1.53$ |
| | Task-Oriented | $75.04 \pm 0.06$ | $81.15 \pm 0.01$ | $84.70 \pm 0.08$ | $87.61 \pm 0.08$ |

Table 1: Classification accuracy on HAM10K dataset.

When performing task-agnostic distillation of frozen VFMs, we use a learning rate of $1e^{-3}$. We train for 100 epochs with a batch size of 2048 on CC3M dataset, 200 epochs with a batch size of 2048 on Places365 dataset, and 10K epochs with a batch size of 1024 on the smaller ADE20K and HAM10K datasets. We also tried longer training runs, but did not see much improvement in the final target task performance. When distilling finetuned VFMs, we use a learning rate of $7e^{-4}$. When distilling VFMs finetuned for Places365 classification, we train for 100 and 200 epochs on CC3M and Places365 datasets, respectively, with a batch size of 512. When distilling VFMs finetuned for HAM10K classification, we train for 100 epochs on CC3M dataset with a batch size of 512 and 7K epochs on the smaller HAM10K dataset with a batch size of 128. When distilling VFMs finetuned for ADE20K segmentation, we train for 15 and 2K epochs on CC3M and ADE20K datasets, respectively, with a batch size of 128.

For downstream finetuning, we tried several learning rates from $7e^{-6}$ to $3e^{-3}$ and report results corresponding to the best ones. When finetuning on Places365 dataset, we use a batch size of 512. We train for 200 epochs when 50 labeled images per class are used, and 100 epochs when 250/1000 labeled images per class are used. When finetuning on HAM10K dataset, we use a batch size of 128 and train for 200 epochs. When finetuning on ADE20K dataset, we use a batch size of 32, and train for 500/400/300/200 epochs when using 1200/2401/4802/9606 labeled images. For each finetuning experiment, we use the corresponding validation dataset to pick the best checkpoint and report results on the test set for this checkpoint.

# B  RESULTS

## B.1  HAM10K CLASSIFICATION RESULTS

Table 1 presents the HAM10K classification results for different combinations of VFM and transfer set with FastViT-S12 as the target architecture. These results correspond to Fig. 3.

## B.2  PLACES365 CLASSIFICATION RESULTS

Table 2 presents the Places365 classification accuracy results for different combinations of VFM, target model and transfer set. These results correspond to Fig. 2.

## B.3  ADE20K SEGMENTATION RESULTS

Table 3 presents the ADE20K segmentation results for different combinations of target model and transfer set with DINOV2 as the VFM. These results correspond to Fig. 4.

| Labeled images per class | | 50 | 250 | 1000 |
|---|---|---|---|---|
| Finetuned DINOV2 | | $47.56 \pm 0.02$ | $54.11 \pm 0.27$ | $56.95 \pm 0.11$ |
| Finetuned OpenCLIP | | $48.30 \pm 0.59$ | $54.26 \pm 0.19$ | $57.03 \pm 0.17$ |
| Target model - MobileViT-V2 | | | | |
| IM-Pretrain | | $40.46 \pm 0.07$ | $47.93 \pm 0.15$ | $52.73 \pm 0.04$ |
| CLIP-Pretrain | | $42.42 \pm 0.05^*$ | $48.06 \pm 0.16$ | $52.72 \pm 0.08$ |
| DINOV2 (CC3M transfer ) | Task-Agn (Patch) | $41.81 \pm 0.16$ | $48.90 \pm 0.12$ | $53.15 \pm 0.08$ |
| | Task-Agn (Image) | $42.55 \pm 0.19$ | $48.89 \pm 0.09$ | $53.05 \pm 0.09$ |
| | Task-Oriented | $47.44 \pm 0.02$ | $52.44 \pm 0.04$ | $54.93 \pm 0.04$ |
| DINOV2 (Places365 transfer) | Task-Agn (Patch) | $45.14 \pm 0.08$ | $50.37 \pm 0.06$ | $53.72 \pm 0.08$ |
| | Task-Agn (Image) | $45.73 \pm 0.02$ | $50.61 \pm 0.05$ | $53.43 \pm 0.03$ |
| | Task-Oriented | $49.20 \pm 0.04$ | $54.02 \pm 0.05$ | $56.10 \pm 0.03$ |
| OpenCLIP (CC3M transfer) | Task-Agn | $43.23 \pm 0.09^*$ | $49.07 \pm 0.04$ | $53.04 \pm 0.14$ |
| | Task-Oriented | $47.82 \pm 0.02$ | $52.82 \pm 0.05$ | $54.92 \pm 0.03$ |
| OpenCLIP (Places365 transfer) | Task-Agn | $47.61 \pm 0.05^*$ | $51.39 \pm 0.10$ | $54.17 \pm 0.11$ |
| | Task-Oriented | $50.14 \pm 0.04$ | $54.30 \pm 0.02$ | $56.43 \pm 0.04$ |
| Target model - FastViT-S12 | | | | |
| IM-Pretrain | | $40.58 \pm 0.14$ | $47.96 \pm 0.09$ | $52.33 \pm 0.14$ |
| CLIP-Pretrain | | $45.17 \pm 0.03^*$ | $47.83 \pm 0.30$ | $52.37 \pm 0.47$ |
| DINOV2 (CC3M transfer) | Task-Agn (Patch) | $42.46 \pm 0.02^*$ | $49.60 \pm 0.11$ | $53.49 \pm 0.06$ |
| | Task-Agn (Image) | $44.52 \pm 0.13^*$ | $49.75 \pm 0.07$ | $53.45 \pm 0.04$ |
| | Task-Oriented | $47.81 \pm 0.05$ | $53.09 \pm 0.05$ | $55.62 \pm 0.05$ |
| DINOV2 (Places365 transfer) | Task-Agn (Patch) | $46.45 \pm 0.05^*$ | $51.26 \pm 0.20$ | $54.43 \pm 0.03$ |
| | Task-Agn (Image) | $47.76 \pm 0.04^*$ | $51.45 \pm 0.32$ | $54.45 \pm 0.21$ |
| | Task-Oriented | $49.14 \pm 0.02$ | $54.51 \pm 0.05$ | $56.68 \pm 0.01$ |
| OpenCLIP (CC3M transfer) | Task-Agn | $45.44 \pm 0.03^*$ | $49.56 \pm 0.05$ | $53.59 \pm 0.09$ |
| | Task-Oriented | $48.75 \pm 0.01$ | $53.33 \pm 0.05$ | $55.75 \pm 0.06$ |
| OpenCLIP (Places365 transfer) | Task-Agn | $48.83 \pm 0.07^*$ | $51.92 \pm 0.26$ | $54.74 \pm 0.10$ |
| | Task-Oriented | $50.47 \pm 0.01$ | $54.82 \pm 0.04$ | $56.80 \pm 0.02$ |

Table 2: Classification accuracy on Places365 dataset. The results marked with $^*$ are obtained by training only the classification layer instead of the entire model in the finetuning stage. Full finetuning produced inferior results in these cases.

### B.4 TRANSFER SET CURATION

Figure 6 compares various transfer sets in terms of the final ADE20K segmentation performance in the context of task-agnostic knowledge transfer between DINOV2 and FastViT-S12 models. Curated transfer sets outperform both ADE20K and CC3M datasets.

### B.5 IMAGE RETRIEVAL

Figure 7 shows few example query images from ADE20K dataset and the corresponding nearest neighbors from YFCC15M dataset.

Figure 8 shows t-SNE visualization of image embeddings of ADE20K dataset and randomly sampled 10% of YFCC15M dataset. Images from ADE20K dataset cover only a small region. This explains the performance drop in Fig. 5 when we retrieve more than 150K images.

| Labeled training images | | 1200 | 2401 | 4802 | 9605 |
|---|---|---|---|---|---|
| Finetuned DINOV2 | | $39.32 \pm 0.11$ | $42.76 \pm 0.08$ | $46.35 \pm 0.09$ | $49.42 \pm 0.14$ |
| Target model - MobileViT-V2 + DeepLab-V3 | | | | | |
| IM-Pretrain | | $22.36 \pm 0.18$ | $26.60 \pm 0.26$ | $30.95 \pm 0.41$ | $35.02 \pm 0.01$ |
| CLIP-Pretrain | | $19.58 \pm 0.22$ | $24.66 \pm 0.09$ | $29.06 \pm 0.24$ | $33.62 \pm 0.14$ |
| DINOV2 (CC3M transfer) | Task-Agn (Patch) | $25.42 \pm 0.18$ | $29.52 \pm 0.10$ | $32.76 \pm 0.28$ | $36.73 \pm 0.17$ |
| | Task-Oriented | $33.35 \pm 0.02$ | $35.68 \pm 0.08$ | $37.32 \pm 0.11$ | $40.14 \pm 0.05$ |
| DINOV2 (ADE20K transfer) | Task-Agn (Patch) | $30.03 \pm 0.15$ | $33.12 \pm 0.27$ | $35.41 \pm 0.15$ | $37.96 \pm 0.39$ |
| | Task-Oriented | $33.19 \pm 0.11$ | $35.01 \pm 0.09$ | $37.33 \pm 0.03$ | $39.07 \pm 0.09$ |
| Target model - FastViT-S12 + DeepLab-V3 | | | | | |
| IM-Pretrain | | $22.58 \pm 0.07$ | $27.11 \pm 0.06$ | $33.12 \pm 0.25$ | $37.54 \pm 0.02$ |
| CLIP-Pretrain | | $15.34 \pm 0.20$ | $20.88 \pm 0.26$ | $27.57 \pm 0.11$ | $32.63 \pm 0.08$ |
| DINOV2 (CC3M transfer) | Task-Agn (Patch) | $25.75 \pm 0.55$ | $30.27 \pm 0.14$ | $35.83 \pm 0.11$ | $39.28 \pm 0.54$ |
| | Task-Oriented | $36.28 \pm 0.01$ | $39.22 \pm 0.19$ | $41.44 \pm 0.15$ | $44.29 \pm 0.03$ |
| DINOV2 (ADE20K transfer) | Task-Agn (Patch) | $28.32 \pm 0.46$ | $32.0 \pm 0.19$ | $36.81 \pm 0.08$ | $39.44 \pm 0.22$ |
| | Task-Oriented | $34.57 \pm 0.02$ | $37.19 \pm 0.10$ | $38.45 \pm 0.05$ | $41.07 \pm 0.21$ |

Table 3: Mean IoU on ADE20K dataset.

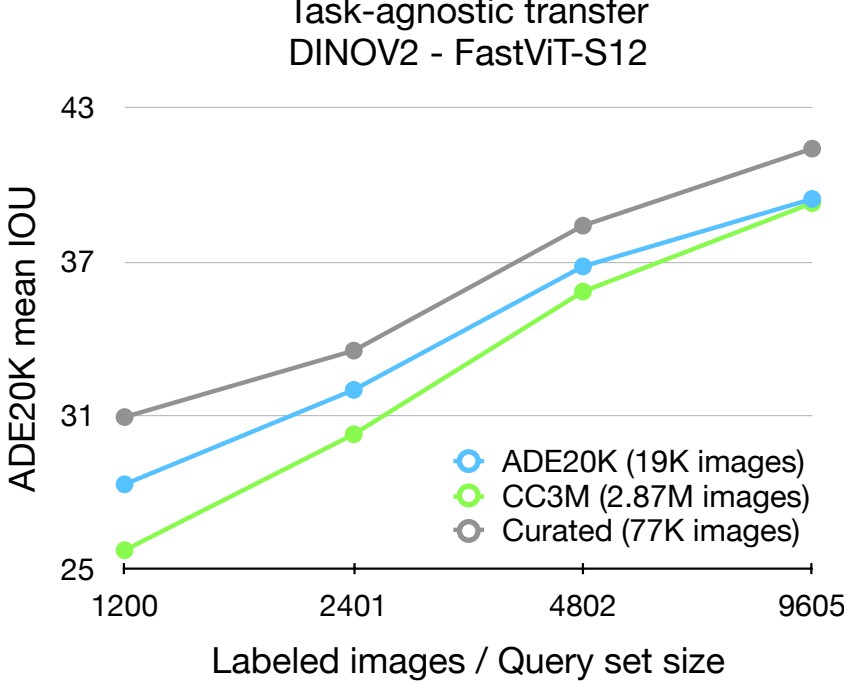

Figure 6: Performance of task-agnostic transfer for various transfer sets. Curated transfer sets outperform both CC3M and ADE20K transfer sets.

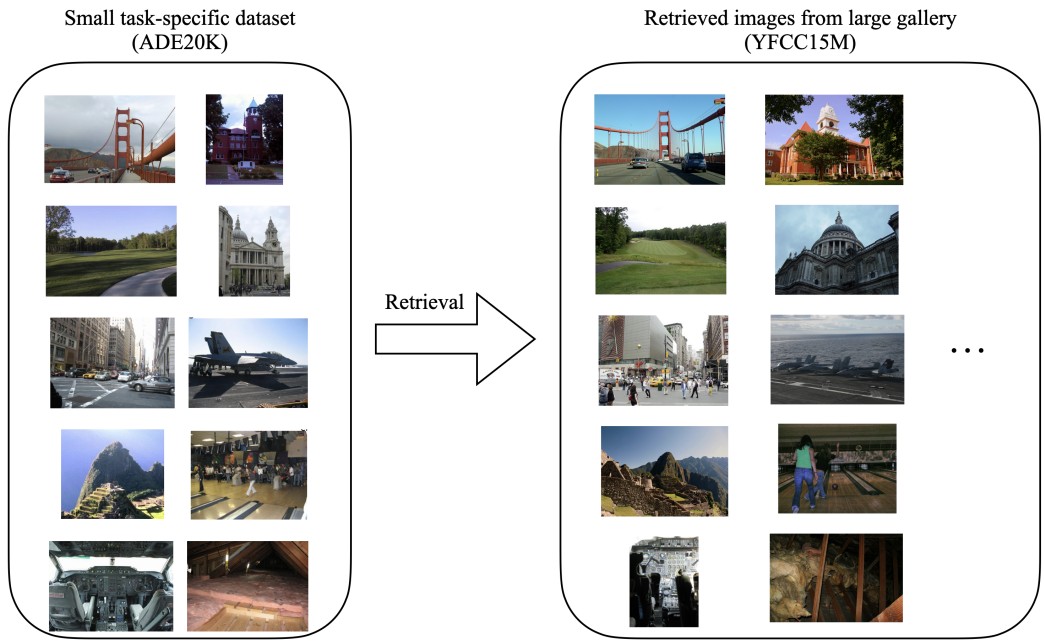

Figure 7: Examples of retrieved images from YFCC15M dataset corresponding to query images from ADE20K dataset.

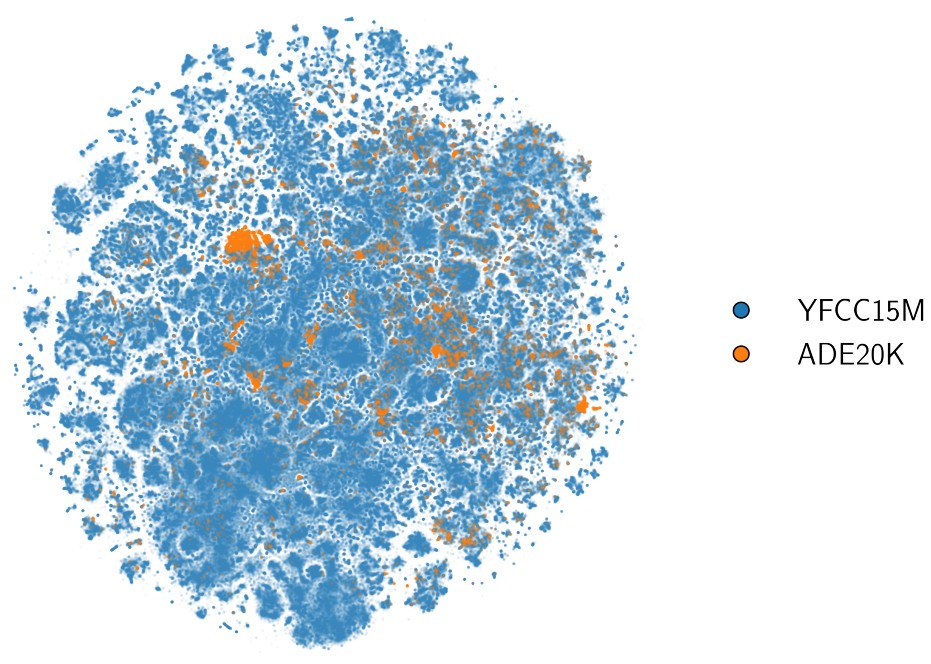

Figure 8: t-SNE (Van der Maaten & Hinton, 2008) visualization of image embeddings of ADE20K dataset and randomly sampled 10% of YFCC15M dataset.

