- **HAM10K skin lesion disease classification** (**?**): This dataset consists of 10K training, 193 validation and 1.5K test images.

- **ADE20K semantic segmentation** (**?**): This dataset consists of 20.2K training and 2K validation images. We split the original training set into two subsets with 19.2K and 1K images, which we use for training and validation, respectively. We report the final evaluation results on the original 2K validation set.

**Evaluation metrics:** We use top-1 accuracy for Places365 and HAM10K classification tasks, and mean Intersection over Union (IoU) for the ADE20K segmentation task.

**Transfer sets**: For each target task, we experiment with two transfer sets. The first one is a generic transfer set consisting of 2.87M unlabeled images from the training split of the CC3M dataset (**?**), and the second one is a task-related transfer set consisting of unlabeled images from the target task domain. For each task, we use the entire training split of the corresponding dataset as the unlabeled task-related transfer set, which contains 1.8M images for Places365 classification, 19.2K images for ADE20K segmentation, and 10K images for HAM10K classification.

**Foundation models:** We use the DINOV2-ViT-L/14 model (**?**) and the OpenCLIP-ViT-L/14 model (**?**) trained on the DataComp-1B dataset (**?**) as VFMs. For brevity, we refer to them as DINOV2 and OpenCLIP, respectively.

**Target models:** We use two recent efficient architectures, namely MobileViT-V2-1.0 (**?**) and FastViT-S12 (**?**), as image encoders for the target models.

**Task-specific heads:** We use a linear classifier as the task-specific head for classification tasks, and a DeepLabV3 head (**?**) as the task-specific head for segmentation tasks. Please refer to Appendix A.1 for further details.

**Loss functions:** For finetuning with labeled target task dataset, we use the standard cross entropy loss, and for matching task predictions, we use KL divergence between the softmax outputs of VFM and target model. For segmentation tasks, these losses are used at each pixel. The loss function used for matching features depends on the VFM. In the case of OpenCLIP model, we use contrastive loss (**?**) with a linear projection layer on top of the target model output to match its dimensionality with the CLIP embedding dimensionality. Since DINOV2 is trained to produce good patch features along with global image features, we experiment with both image-level and patch-level features in the case of DINOV2. When using global image features for knowledge transfer, we use contrastive loss with linear projection layers on outputs of both models. When using patch features for knowledge transfer, we use cosine similarity loss with a linear projection layer on top of the target model features for dimensionality matching. We also resize DINOV2 patch features to match the spatial resolution of the target model features.

In each experiment, we run the finetuning step three times and report the average results.

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

- **Random:** Randomly select images from the gallery.

We use the DINOV2 foundation model and FastViT-S12 target model for these experiments.

Figure 5 (left) shows the final ADE20K segmentation performance for task-agnostic transfer using transfer sets curated by different retrieval strategies. Here, we use 4800 labeled images for finetuning the target model and use the same 4800 images as the query set for retrieval. Query-balanced retrieval based on image queries performs the best. By giving equal weight to each query, this approach increases diversity in the retrieved samples when compared to the best-matches strategy. The segmentation performance increases as we increase the transfer set size until we reach 77K-154K images and starts to drop slowly after that. We performed a similar experiment for task-oriented transfer using the best performing image query-balanced retrieval strategy. Figure 5 (middle) shows the corresponding results. Again, the performance steadily increases until we reach 154K images and starts to drop after that indicating that the size of YFCC15M subset that is most useful for ADE20K segmentation is around 154K.

Using 154K as the target transfer set size, we curated different transfer sets by varying the number of query images used for retrieval. Figure 5 (right) compares the performance of these curated transfer sets with the CC3M and ADE20K transfer sets for task-oriented transfer. Curated transfer sets clearly outperform the smaller task-related ADE20K transfer set and the much larger generic CC3M transfer set. [2]

## 4 RELATED WORKS

**Knowledge distillation** is a widely-used approach for transferring knowledge between model architectures by training one model to mimic the outputs of another model. Numerous knowledge distillation approaches have been proposed over the past decade based on various knowledge representations such as task logits (**?**), intermediate features or embeddings (**??**), relations between

---

[2]The best performance in the middle figure is lower than the performance for curated transfer set corresponding to 4800 query images in the right figure. This is because, we used shorter training runs (60K steps) to get the results in the middle figure, and once we identified the best transfer set size, we used longer training runs (200K steps) for the right figure.

samples (**?**), attention maps (**?**), etc. Please refer to **??** for a comprehensive survey of existing knowledge distillation approaches.