# OpenReview forum: "Label-efficient Training of Small Task-specific Models by Leveraging Vision Foundation Models"
_ICLR.cc/2024/Conference — Submitted to ICLR 2024_

### Official Review · Reviewer_Ph8P · 2023-10-22

**Soundness:** 3 good
**Presentation:** 3 good
**Contribution:** 3 good
**Rating:** 5
**Confidence:** 4

**Summary:**

The paper presents a method called "task-oriented knowledge transfer" for training smaller, task-specific models by leveraging knowledge from large Vision Foundation Models (VFMs). This approach involves three steps:

1. Fine-tuning the VFM on the target task using labeled data.
2. Transferring knowledge from the fine-tuned VFM to the target model using an unlabeled dataset (transfer set) based on the knowledge distillation framework.
3. Fine-tuning the target model with labeled target task data.

An alternative method, called "task-agnostic knowledge transfer", involves distilling knowledge from the frozen VFM image encoder to the target model image encoder and then fine-tuning the target model using labeled data.

The paper makes a comparison of task-oriented knowledge transfer with task-agnostic transfer, direct web-scale CLIP pretraining, and supervised ImageNet-1k pretraining. The authors show that the task-oriented knowledge transfer method performs better on many benchmarks.

The study also emphasizes the importance of the transfer set, where using a transfer set with an image distribution similar to the target task image distribution leads to better performance. For cases where a large target task-related transfer set is not readily available, the paper proposes a solution: curating task-related transfer sets using image retrieval with images from the limited labeled target task dataset as queries. Authors show that this method improved segmentation performance when compared to using a generic transfer set.

**Strengths:**

Originality:
- Proposes a new task-oriented knowledge transfer approach to leverage large pretrained VFMs to train small specialized models for new tasks with limited labeled data.
- While retrieval-based strategies have been explored before, using retrieval to curate task-related transfer sets specifically for knowledge transfer is a creative application.
However, the core ideas of knowledge distillation and leveraging related datasets are not entirely new. The novelty lies in how these techniques are tailored and applied.

Quality:
- Comprehensive experiments comparing task-oriented transfer to baselines like task-agnostic transfer, CLIP pretraining, and ImageNet pretraining.
- Also has an ablation studies analyzing impact of transfer set distribution.

Clarity:
- The comparisons to baselines and ablation studies are well-presented.
- The limitations of the approach are clearly stated.

Significance:
- Enables leveraging powerful VFMs for specialized small model training under limited target data regimes.
- Shows the potential to learn specialized models even for domains not well-covered by web data.

**Weaknesses:**

- The curated task-related transfer sets are shown to be effective, but the retrieval approach to create them is rather simple. There is scope to explore more sophisticated retrieval and data selection techniques like active learning, core-set selection, adversarial filtering etc. This could potentially lead to even better task-specific transfer sets.
- The paper could benefit from a comparative analysis with a few other knowledge transfer approaches. This would help provide a clearer picture of how the proposed method stands against existing techniques.

**Questions:**

- The paper mentions that the target models may inherit the biases of the foundation models. It would be beneficial to discuss potential ways to address this issue.
- Lack of detail on fine-tuning VFMs: The paper mentions using labeled target task data to fine-tune VFMs but doesn't provide much detail on this process. More information on how VFMs are fine-tuned could help readers better understand the complete methodology.

---

> ### Author Response · Authors · 2023-11-11
> **Thank you for your constructive feedback.**
>
> We thank the reviewer for their positive and constructive comments, and address their concerns below. We hope that this rebuttal addresses all their questions and convinces them to raise their ratings. Please let us know if any additional details/clarifications are needed.
>
> **Sophisticated curation approaches:**
> The main focus of this paper is on leveraging VFMs to effectively pretrain small task-specific models, not the specific strategy used for curating the transfer set. We would like to highlight that, in all our experiments, pretraining using VFMs is better than ImageNet/CLIP pretraining even when generic CC3M dataset is used for knowledge transfer. So, it is not absolutely necessary to use task-related transfer set to outperform ImageNet/CLIP pretraining. Using a task-related transfer set just makes the performance gap even bigger. Our results on ADE20K dataset show that a simple image retrieval approach can also be effective in curating good transfer sets. We argue that **simplicity** should be considered as one of the strengths of our approach. We agree with the reviewer that a more sophisticated curation process could further improve the results, and we plan to explore this direction in our followup works.
>
> **Additional comparisons:**
> ImageNet pretraining and CLIP pretraining are the most popular and widely-used pretraining approaches in the context of transfer learning. Hence, we used them for comparison in this paper. To be more comprehensive, we performed additional experiments comparing the proposed approach with an alternative transfer learning strategy: self-supervised pretraining followed by finetuning. Please see the results here: https://openreview.net/forum?id=HnVtsfyvap&noteId=7vBTW2DhMA
>
> Our main goal is to establish “Adapt VFM → Transfer knowledge across architectures → Finetune small model” as a highly effective strategy for training small task-specific models. In this work, we used supervised finetuning strategy for adapting VFMs and feature/task-prediction matching for transferring knowledge across architectures, and showed that we already outperform other widely-used approaches. More sophisticated finetuning/distillation approaches can easily be incorporated into our general strategy to further improve the performance.
>
> **Details of finetuning VFMs:**
> The process of finetuning VFMs is similar to the process of finetuning small models. We attach a randomly initialized task-specific head to a pretrained VFM encoder, and finetune the entire model. The details of task-specific heads are described in Appendix A.1. The optimizer,  learning rates, number of epochs and image augmentations used for finetuning VFMs are same as the ones used for finetuning small models. Please see Appendix A.2 for details. We will make these details clearer in the final draft. Please let us know if there are any specific details that we may have missed.
>
> **Biases:**
> A VFM pretrained on web data may have its own inherent biases that may be useful/harmful for a particular target task. The proposed approach advocates for adapting pretrained VFMs to target task/domain before transferring their knowledge to small models. This strategy gives us an opportunity to enhance/suppress desired/irrelevant knowledge in VFMs by using appropriate target domain data. Using a well-curated and balanced target domain set for knowledge transfer can also help us mitigate some of the biases. This is an interesting line of research for followup works.

---

> > ### Comment · Reviewer_Ph8P · 2023-11-22
> > **Thank you for the follow up answer.**
> >
> > I appreciate the author's answers. My rating stays the same.

---

### Official Review · Reviewer_H2Cx · 2023-10-26

**Soundness:** 4 excellent
**Presentation:** 3 good
**Contribution:** 3 good
**Rating:** 6
**Confidence:** 4

**Summary:**

This work study the problem of knowledge transfer from vision foundation model. Specifically, VFM are use to train a small task-specific dataset with limited labeled training data.

**Strengths:**

+ Proposed a three stages training strategy to train small DNN models with limited labeled training data by utiziling knowledge from VFMs. The VFM is first finetuned with the target task data and task-specific head, hence the finetuned CFM is more suitable for transfers task-oriented knowledge to the target small model with a transfer set.
+ This work observe that the transfer dataset distribution play an important role on the knowledge transfer, and empirically validate several approaches to curate the transfer dataset.
+ The resulting models (demonstrated with two VFMs and two mobile target architectures) show good performance than task-agnostic VFM distillation or pre-trained models. The empirical results support several insights that are benefitial to the research community.

**Weaknesses:**

This work show a task-oriented knowledge transfer strategy. The method is simple and intuitive. One might be complaining all the computation tools are from prior art, and the technical novelty is low. But i think such work bring good contribution to the research community, to more effectively train specialist small model. I don't see major flow on the proposed approach.

**Questions:**

On Fig 2(a), what are the three lines for DINOV2 - MobileViT-V2 - CC3M?

Note: the uploaded supplementary material is almost identical with the main paper, but with broken reference in text.

---

> ### Author Response · Authors · 2023-11-11
> **Thank you for the positive review.**
>
> We sincerely thank the reviewer for their positive feedback and for explicitly calling out that this work brings good contributions to the research community. We believe that **simplicity** is one of the greatest strengths of our approach and we appreciate the reviewer for supporting us on this. We hope that this rebuttal addresses all their questions and convinces them to raise their ratings. Please let us know if any additional details/clarifications are needed.
>
> **Fig. 2(a):**
> Sorry to say that we did not understand your question fully. The red curve in all the plots corresponds to fine-tuned VFM. Hence, when the VFM is DINOV2, the red curves correspond to fine-tuned DINOV2 (not fine-tuned OpenCLIP). We will add a separate legend for plots corresponding to DINOV2 to clarify this. After the submission, we learned that some of the plots are not showing up correctly in some viewers. If this the case, please view the paper using Adobe/Chrome.
>
> **Supplementary material:**
> Thank you for pointing this out.  A copy of the main paper was uploaded as supplementary material by mistake.
>
> **New additional results:**
> To further strengthen the paper, we conducted additional experiments. Please see https://openreview.net/forum?id=HnVtsfyvap&noteId=7vBTW2DhMA

---

> > ### Comment · Reviewer_H2Cx · 2023-11-23
> >
> > I acknowledge the author rebuttal. I will consider the rebuttal and fellow reviewers' opinion for final rating.

---

### Official Review · Reviewer_KUqe · 2023-10-26

**Soundness:** 3 good
**Presentation:** 3 good
**Contribution:** 3 good
**Rating:** 6
**Confidence:** 3

**Summary:**

This paper studies how we can train a smaller model for a new target task with limited training data using vision foundation models. They propose two frameworks task-agnostic and task-oriented for distilling data from foundation model to smaller model. Task agnostic or task oriented knowledge distillation involves creation of target set to transfer information from vision foundation model to smaller model. They show that their task agnostic approach or task oriented approach beats the models pretrained on large datasets like Imagenet or web-scale CLIP pretraining.

**Strengths:**

- Simple and clean experiments. Paper is well written as well. They experimented with different datasets including imabalnced ones showing the effectiveness of their approach.
- The paper shows that task agnostic and task oriented knowledge transfer beats the imagenet pretraining and CLIP based pre-training.

**Weaknesses:**

- Most of the findings in the paper are obvious like task oriented or task agnostic knowledge transfer would lead to better performance than model pre-trained on CLIP or Imagenet because we are creating transfer set similar to target dataset.
- Transfer set creation process is mostly heuristic based and it's one of the important factors to get good performance on the target task.

**Questions:**

- Based on the figure 5 (left) if we use random transfer set the performance of the model on target task is similar to models pretrained on Imagenet or CLIP based pretraining.
- Additionally, it's essential to clarify whether the target models, specifically MobileViT-v2, are pre-trained. If they are indeed pre-trained, the comparison between a 'normal' pre-trained model and a fine-tuned model (IM-Pretrain, CLIP-Pretrain), versus a model that undergoes pre-training followed by further fine-tuning with task-specific or task-agnostic training (Task-Agn, Task-Oriented), may lack meaningful context.

---

> ### Author Response · Authors · 2023-11-11
> **Thank you for the positive review.**
>
> We sincerely thank the reviewer for their positive feedback and address their concerns below. We hope that this rebuttal addresses all their questions and convinces them to raise their ratings. Please let us know if any additional details/clarifications are needed.
>
> **Obvious findings:**
> As far as we know, there is no existing work that concretely establishes the findings presented in this paper about leveraging VFMs for training mobile models on a new task with limited labeled data. Especially, we show that
>
> 1. Knowledge transfer (both task-oriented and task-agnostic) from a VFM (even with generic CC3M transfer set) is a better pre-training strategy for small models when compared to supervised ImageNet or web-scale CLIP pretraining.
> 2. Task-oriented knowledge transfer from VFMs is better than task agnostic transfer.
>
> The reviewer seems to be under the impression that proposed knowledge transfer outperforms ImageNet/CLIP pretraining because we are curating transfer sets. We would like to emphasize that knowledge transfer outperforms ImageNet/CLIP pretraining even when using the generic CC3M transfer set. A curated transfer set just makes the performance gap even bigger. We strongly believe that these non-trivial findings corroborated by our thorough experimental validation would be very useful to the community, especially given that VFMs are becoming larger and difficult to deploy as time progresses.
>
> **Transfer set creation:**
> The main focus of this paper is on leveraging VFMs to effectively pretrain small task-specific models, not the specific strategy used for curating transfer set. Our results on ADE20K dataset show that a simple image retrieval approach can also be effective in curating good transfer sets. We agree that a  more sophisticated curation process could further improve the results. However, that is not the main focus of this paper and we plan to explore this direction in our followup works.
>
> **Random transfer set performance:**
> Random transfer set performance in Fig. 5(a) is not similar to ImageNet/CLIP pretraining as claimed by the reviewer. It indeed outperforms ImageNet/CLIP pretraining. Below are the ADE20K segmentation performances when using 4800 labeled images:
>
> | Fig. 5(a) - Random subset of YFCC (154K images) | 35.7 |
> |---|---|
> | Table 3 - ImageNet pretraining | 33.1 |
> | Table 3 - CLIP pretraining | 27.6 |
>
>
> **Additional pretraining:**
> When using the proposed knowledge transfer approaches (task-oriented or task-agnostic), target models are trained from scratch. We agree that starting knowledge transfer process from an already pretrained model would be an unfair comparison. We did not do that. We will make this clear in the final draft.
>
> **New additional results:**
> To further strengthen the paper, we conducted additional experiments. Please see https://openreview.net/forum?id=HnVtsfyvap&noteId=7vBTW2DhMA

---

### Official Review · Reviewer_M3yw · 2023-10-30

**Soundness:** 2 fair
**Presentation:** 2 fair
**Contribution:** 1 poor
**Rating:** 3
**Confidence:** 4

**Summary:**

This paper investigates how to construct a compute-efficient model from a pre-trained vision model for a downstream task with limited labeled training examples. To approach the task, the authors propose to first fine-tune the pre-trained vision model with the labeled data, distill it into a smaller model with a transfer set, and then fine-tune the distilled model with the labeled dataset. The effects of using in-domain and out-of-domain transfer sets are investigated and experiments are conducted to show the superiority of the approach against multiple baselines.

**Strengths:**

1. The investigation of using non-in-domain data for distillation and selecting relevant data for distillation is interesting.

**Weaknesses:**

1. Lacking Novelty: In essence, the proposed approach is knowledge distillation. Under the setup the authors consider, the proposed approach is the most straightforward approach one could consider. While the investigation of the use of not in-domain transfer dataset is interesting, the finding — when sufficient in amount, in-domain transfer set is better — is somewhat expected (e.g prior work[1] on open-set semi-supervised learning has shown that out-of-domain example can hurt performance of semi-supervised algorithm). It would be more interesting if the authors investigated the use of data from different domains (medical, satellite, clipart, etc) for distillation and proposed a strategy for selecting the right data for distillation. The current submission does contain some investigation on selecting the right data for distillation (section 3.5), but the investigation is not thorough enough to convince the reviewer that their strategy is applicable for different target tasks and with data from different domains.
2. Problematic Baselines: One would consider using the task-agnostic approach if fine-tuning the pre-trained models could be an issue (perhaps due to compute-constraint) but since the authors only consider compute-constraint during inference (instead of training), it seems unnatural to consider task-agnostic distillation as a baseline. In addition, the CLIP-pretrain baseline is trained on an internal dataset. Without knowing the distribution/statistics of the data and how the dataset compares to CC3M or DataComp-1B, it is difficult to understand whether the CLIP-pretrain baseline is worse because contrastive pretraining is not the right approach or the data used for training is flawed

Citation:
[1] Saito, Kuniaki, Donghyun Kim, and Kate Saenko. "Openmatch: Open-set semi-supervised learning with open-set consistency regularization." Advances in Neural Information Processing Systems 34 (2021): 25956-25967.

**Questions:**

Questions:
1. 3.1 Alternative Approaches CLIP-Pretrain: The authors mentioned using a loss function similar to CLIP. Could the authors clarify what the difference is?
2. Have the authors tried distilling from the patch tokens from CLIP?


Suggestions:
1. Additional baseline: CLIP is a weakly supervised pre-training approach, the reviewer recommends adding self-supervised pre-training approaches such as DINO.
2. Transfer set: The use of not in-domain transfer set is worth more explanation. Right now, there is not much explanation/description on the use of not in-domain transfer set (e.g. why an in-domain transfer set is difficult to obtain).
3. Distillation mechanism: Distilling patch features is important for the segmentation task. However, there are only a few sentences describing the process. The reviewer recommends expanding the section on loss function either via using equations (the use of contrastive loss for distillation is uncommon) or by providing a figure explaining the process.

Pre-rebuttal rating: Overall, the paper contains some interesting investigations on how to use distillation for constructing a small model from a large model pre-trained on different source tasks. However, the investigations are so far incomplete and the quality is not up to par with ICLR’s standards. The reviewer thus recommends rejecting the submission.

**Details Of Ethics Concerns:**

NA.

---

> ### Author Response · Authors · 2023-11-11
> **Thank you for your constructive feedback.**
>
> We thank the reviewer for their constructive suggestions and address their concerns below. We hope that this rebuttal addresses all their questions and convinces them to raise their ratings. Please let us know if any additional details/clarifications are needed.
>
> **Novelty:**
> As far as we know, there is no existing work that concretely establishes the findings presented in this paper about leveraging VFMs for training small models on a new task with limited labeled data. Especially, we show that
>
> 1. Knowledge transfer (both task-oriented and task-agnostic) from a VFM (even with generic CC3M transfer set) is a better pre-training strategy for small models when compared to ImageNet/CLIP pretraining.
> 2. Task-oriented knowledge transfer from VFMs is better than task-agnostic transfer.
>
> We strongly believe that our findings corroborated by thorough experimental validation would be very useful to the community, especially given that VFMs are becoming larger and difficult to deploy as time progresses.
>
> The main focus of this paper is on leveraging VFMs to effectively pretrain small task-specific models, not the specific strategy used for curating the transfer set. Please note that knowledge transfer is better than ImageNet/CLIP pretraining even when generic CC3M transfer set is used. So, it is not necessary to use task-related transfer set to outperform ImageNet/CLIP pretraining. Using a task-related transfer set just makes the performance gap bigger. Our results on ADE20K show that a simple image retrieval approach can be effective in curating good transfer sets. While a more sophisticated curation process could further improve the results and **transfer set curation** is an interesting topic worth exploring in further detail, it is not the main focus of this work. However, we agree that it would be an excellent topic for followup works.
>
> **Baselines:**
> Sorry to say that we did not understand why task-agnostic distillation is “unnatural”.  Task-agnostic distillation is a valid approach to leverage VFMs, and hence we compared it with the proposed task-oriented knowledge transfer in a fair manner using same amount of labeled and unlabeled data.
>
> **CLIP dataset:**
> We use the 1.1B image-text pairs used in the following papers: “STAIR: Learning Sparse Text and Image Representation in Grounded Tokens” and “RangeAugment: Efficient Online Augmentation with Range Learning”.
> ViT-B/16 and ViT-H/16 CLIP models trained on this dataset achieve 72.8% and 78.6% ImageNet zero-shot accuracy which are comparable to OpenCLIP models trained on DataComp-1B. These results confirm that the dataset we use is a high quality dataset.
>
> **CLIP-Pretrain:**
> Sorry for the confusion. Our loss function is same as the loss function in the original CLIP paper. By ‘similar’ we simply meant ‘same’.
>
> **Distilling patch tokens from CLIP:**
> We tried this in our early experiments but did not observe promising results. DINOV2 patch features are significantly better than OpenCLIP patch features for segmentation. This is because DINOV2 has been explicitly trained to produce good patch features while OpenCLIP was trained only with a loss on CLS features. Below are the performances of OpenCLIP-ViT-L/14 and DINOV2-ViT-L/14 models on the ADE20K segmentation dataset with Deeplab-V3 segmentation head:
>
> | Labeled images | 1200 | 2401 | 4802 | 9605 |
> |---|---|---|---|---|
> | OpenCLIP | 31.82 | 36.69 | 40.22 | 42.85 |
> | DINOV2 |  39.32 | 42.76 | 46.35 | 49.42 |
>
> DINOV2 outperforms OpenCLIP by a significant margin. Hence, we did not further explore the direction of distilling inferior CLIP patch features. The following paper also observed that patch features of CLIP models trained with CLS token are not good for dense predictions tasks: “Perceptual Grouping in Contrastive Vision-Language Models”.
>
> **Comparison with DINO:** Please see https://openreview.net/forum?id=HnVtsfyvap&noteId=7vBTW2DhMA
>
> **Not in-domain transfer set:**
> If we have large enough task-related data, we should always use that as transfer set. However, the amount of readily available task-related data depends on the domain/task. For example, the number of images in AD20K dataset is around 20K which is not large enough as demonstrated by our experiments. We get better results by using not in-domain CC3M transfer set instead of the limited in-domain ADE20K dataset (see Fig. 4). We will make this more clear in the final draft.
>
> **Distillation loss:**
> We would like to clarify that we do not use contrastive loss for distilling patch features. As mentioned in Sec 3.1 (towards the end of the paragraph on loss function), we use standard cosine similarly loss for patch feature distillation. Contrastive loss is used when distilling global image features following the popular contrastive representation distillation approach. Yonglong Tian, Dilip Krishnan, Phillip Isola, “Contrastive Representation Distillation”, ICLR 2020. We will describe the loss functions in further detail in the final version.

---

> > ### Comment · Reviewer_M3yw · 2023-11-19
> > **Responses to the rebuttal**
> >
> > The reviewer thanks the authors for the detailed responses and clarifications!
> >
> > Re Novelty:
> > 1. [1] considers training small models with limited labeled data using large pre-trained models. The differences in setup between [1] and the submission are (a) multiple pre-trained models are available and (b) no fine-tuning of the pre-trained models is allowed. The reviewer would argue that the setup considered in [1] is much more realistic since with the convenience of huggingface, multiple pre-trained models are accessible; it is unclear why one would restrict themselves to just a single pre-trained model as considered in the submission. Besides, the premise of the work is to construct compute-efficient models. Extrapolating from the development of large models in NLP plus the current rate of development for vision frontier models [2], the reviewer would imagine that next-generation frontier models would be much bigger than what we have right now --- so big that it might not even be possible for a lot of organizations/users to even fine-tune these models (e.g GPT-4) without reliance on corporate-level computes. The fact that the proposed approach relies on fine-tuning the pre-trained models first seems restrictive, assuming we are gonna get larger frontier models.
> > 2. Also, similar results on knowledge distillation/transfer outperforming ImageNet pretraining as also been reported [1]. It is not surprising that similar results would hold for CLIP. The use of a generic CC3M transfer set is interesting. Still, given that both the transfer set and most downstream tasks are internet imagery, the results are again not beyond expectation.
> > 3. task-oriented vs task-agnostic: the pre-trained models are already fine-tuned to be more well-suited for the downstream tasks so it is not beyond expectation that the task-oriented approach is much better.
> >
> > Re Baselines:
> > 1. The reviewer apologizes for the poor choice of words. Instead of "unnatural", "unfair" would be the more fitting option.
> > 2. The reviewer thinks the task-agnostic distillation is unfair since the task-oriented approach leverages fine-tuning which would consume much more compute than the task-agnostic approach. The reviewer cannot help but wonder: with the same amount of compute, could the task-agnostic approach outperform the task-oriented approach? For instance, with the same amount of compute, one could construct an ensemble of downstream models and then further distill this ensemble into a single model. Prior work has shown that distilling an ensemble can yield more performant models[3] and it is unclear whether distilling an ensemble would be better than the proposed task-oriented approach.
> > 3. Also, since supervised fine-tuning is on the table, why not consider semi-supervised fine-tuning as well?
> >
> >
> >
> >
> >
> > References:
> >
> > [1] Borup, Kenneth, Cheng Perng Phoo, and Bharath Hariharan. "Distilling from Similar Tasks for Transfer Learning on a Budget.". In Proceedings of the IEEE/CVF International Conference on Computer Vision, pp. 11431-11441. ICCV, 2023.
> >
> > [2] Dehghani, Mostafa, Josip Djolonga, Basil Mustafa, Piotr Padlewski, Jonathan Heek, Justin Gilmer, Andreas Peter Steiner et al. "Scaling vision transformers to 22 billion parameters." In International Conference on Machine Learning, pp. 7480-7512. PMLR, 2023.
> >
> > [3] Malinin, Andrey, Bruno Mlodozeniec, and Mark Gales. "Ensemble distribution distillation." arXiv preprint arXiv:1905.00076 (2019).
> >
> > [4] Abuduweili, Abulikemu, Xingjian Li, Humphrey Shi, Cheng-Zhong Xu, and Dejing Dou. "Adaptive consistency regularization for semi-supervised transfer learning." In Proceedings of the IEEE/CVF conference on computer vision and pattern recognition, pp. 6923-6932. 2021.

---

> > > ### Comment · Reviewer_M3yw · 2023-12-01
> > > **Post Rebuttal**
> > >
> > > The reviewer apologizes for the delay!
> > >
> > > General comment: The contribution of this paper is a combination of a novel practical setup (transfer learning with compute constraint) and a simple task-specific distillation strategy for the setup. On the setup side, the main attraction is practicality. However, the paper fails to consider the existence of multiple pre-trained models and training compute, diminishing the practicality of the setup. On the method side, the proposed approach requires fine-tuning large VFMs which could require high-end GPUs with large VRAMs and more FLOPs. None of the competing baselines are constructed using a similar amount of FLOPS/VRAMs, thus it is difficult to judge under the same amount of compute budget whether the proposed method is ideal or one could come up with other methods such as distilling an ensemble of compute-efficient models. The paper does contain an interesting finding: an internet dataset CC3M could be used for distilling a model for a non-internet application (HAM10k). However, with only a single instance of success, it is difficult to conclude that the same conclusion would hold for more non-internet applications. Given these considerations, the reviewer decided to keep the rating as is.
> > >
> > >
> > > Below are some specific comments:
> > > 1. Comparison with CLIP: It is not apriori clear that a model trained on a larger dataset would transfer better. Prior investigation on transferability [1, 2] has suggested that the performance of the downstream task depends on the similarity between the downstream task and the pre-training tasks. In this case, the results we observe in the paper might be due to a closer match between ImageNet data and the downstream tasks. Right now, we could only reasonably conclude that the CLIP model does not work well for the 3 tasks that are considered, but it is not clear whether the conclusion would hold for other tasks from different domains (e.g satellite imagery, microscopic images, etc)
> > >
> > > 2. Works without target domain unlabeled data: The reviewer agrees that this finding is interesting. However, it is difficult to deduce whether a similar conclusion would hold in general since 2 out of 3 downstream tasks are natural images (similar to CC3M thus difficult to say whether the unlabeled data is out-of-domain). While HAM10k is different from typical internet imagery, it is difficult to conclude whether using CC3M would be effective for other non-natural image domains such as satellite imagery.
> > > 3. Comparison to Borup et al: Yes, Borup et al do not use the current generation of VFMs but they focus on using a diverse set of pre-trained models. One can extend their work by adding the current generation of VFM to the mix and their conclusion should still be applicable. Also, they do use classification models for experimentation but one could extend their approach by changing the supervised loss to whatever loss for the target task and distillation loss to distill feature vector instead of pseudolabels. The reviewer is concerned that if we accept that the differences between tasks and pre-trained models are major enough to warrant acceptance, then we might end up having needed to accept similar papers when new generations of VFMs come out.
> > > 4. Training compute: GPU hours are only one aspect of compute. The amount of VRAMs and FLOPS required to fine-tune the pre-trained VFMs should not be ignored. Access to high-end GPUs with large VRAMs is not universal --- not everyone has access to high-end GPUs. While PEFT methods such as LoRA could be a solution, it comes with performance degradation. In the setup we consider for this paper, it is unclear how detrimental the performance degradation brought by PEFT would affect the task-specific distillation.
> > >
> > >
> > >
> > > References:
> > >
> > > [1] Bolya, Daniel, Rohit Mittapalli, and Judy Hoffman. "Scalable diverse model selection for accessible transfer learning." Advances in Neural Information Processing Systems 34 (2021): 19301-19312.
> > >
> > > [2] Zamir, Amir R., Alexander Sax, William Shen, Leonidas J. Guibas, Jitendra Malik, and Silvio Savarese. "Taskonomy: Disentangling task transfer learning." In Proceedings of the IEEE conference on computer vision and pattern recognition, pp. 3712-3722. 2018.

---

> ### Author Response · Authors · 2023-11-19
> **Additional response to reviewer M3yw**
>
> We would like to thank the reviewer again for their time and feedback. We hope that this response address their concerns and they would reconsider their score.
>
> We appreciate the reviewer's effort in bringing [1] to our attention. We will include related discussion in our revised version. Similar to our work, [1] focuses on learning a small task-specific model with limited labeled data, and shows that knowledge distillation from the right teacher model (the focus of their work) is better than ImageNet pretraining. However, there are many differences between our work and [1], and our work conveys several important and novel messages that are missing in [1]:
>
> **Comparison with CLIP**  [1] does not compare with CLIP training. We show that knowledge transfer from VFMs outperforms CLIP training. We would like to respectfully disagree with the reviewer comment "It is not surprising that similar results would hold for CLIP". None of the existing works show how effective web-scale CLIP pretraining is for transfer learning in the context of small models. In fact, one would have expected that due to the scale of data, CLIP pretraining may outperform ImageNet pretraining and may even be competitive with knowledge transfer. **Our experiments surprisingly show the opposite trends**.
>
> **Works without target domain unlabeled data** [1] assumes the availability of target domain unlabeled dataset. In contrast, we show that even when target-task unlabeled data is unavailable, knowledge transfer from VFMs using generic internet imagery is highly effective. We think **this is interesting and beyond expectation**. Note that HAM10K data is quite different from typical internet imagery and still knowledge transfer using CC3M dataset outperforms other popular pretraining strategies.
>
> **Experiments with VFMs**
> Note that [1] does not experiment with VFMs. Their teacher models are relatively small CNNs trained on small-scale classification datasets. So, the utility of [1] is unclear in the context of large scale VFMs trained on web-scale datasets. Different from [1], we experiment with VFMs and present many interesting results.
>
>  **Transfer set curation** We a present a simple and effective transfer set curation method using retrieval, and further demonstrate the importance of a balanced curation with respect to seed query set (Figure 5). In contrast, [1] assumes that a large target domain unlabeled dataset is always available.
>
> **Knowledge transfer formulation** [1] assumes that source models are classification models. They use a distillation loss that matches the target and source model predictions in the label space of source model (see Eq. (3) in [1]).  Unfortunately, most of the existing VFMs (CLIP, DINOV2, SAM, etc.) are not supervised classification models, and the formulation of [1] is not directly applicable to them. One could extend [1] to VFMs by using feature matching instead of source label matching as the loss function. Then, it becomes similar to our task-agnostic transfer. As shown in our results, task-agnostic transfer is inferior to task-oriented transfer.
>
> **Works well for segmentation task** [1] only experimented with classification tasks. We show the effectiveness of knowledge transfer from VFMs for semantic segmentation task.
>
> **Comparison with SSL** [1] does not compare with the DINO, which is a popular SSL pretraining approach. Our new results show that knowledge transfer from VFMs is significantly better than DINO pretraining.
>
>
> **Training compute** Note that our work focuses on inference time compute constraints and hence on small task-specific models. While experimenting under a fixed training budget is also an interesting problem, that is not the focus of this work. That said, finetuning the VFMs we used in this work is not as costly as the reviewer might be thinking. The key thing to notice here is that **VFMs are fine-tuned only on a small labeled dataset**. The actual compute used for finetuning VFM is a small fraction of the compute used for distillation in our experiments. For example, finetuning OpenLCIP-ViT-L on Places365 dataset with 50 images/class for 200 epochs takes around 16 A100 GPUh, and distilling frozen OpenCLIP model on full unlabeled Places dataset for 200 epochs takes around 370 A100 GPUh. Also, as VFMs become larger and larger in the future, one could use efficient fine-tuning approaches, such as LoRa[2], which have been shown to be quite effective in the NLP community.
>
>
> **Using multiple models and semi-supervised fine-tuning**  We agree that we can further improve the proposed approach by semi-supervised finetunig of VFMs or by using multiple VFMs. We will look into these directions in the followup works. However, this **doesn’t change the significance of any of the conclusions presented in this paper**. Note that, while finetuning a VFM on a small labeled dataset is computationally inexpensive, finetuning it in a semi-supervised fashion on large unlabeled dataset may not be.

---

### Author Response · Authors · 2023-11-11
**Additional results to further strengthen the paper**

We thank all the reviewers for their constructive feedback.

Reviewers seem to agree that the proposed knowledge transfer approach is highly effective while being simple, our experimental analysis is thorough and provides some interesting insights. Given its strong results, we argue that **simplicity** of our approach should be seen as one of its greatest strengths since simple approaches are often the ones that are widely adopted by the community.

To further strengthen the paper, we present the following additional results in this rebuttal:

1. Results on ImageNet dataset as a fourth target task (in addition to Places365, ADE20K and HAM10K) - Similar to the other three target tasks, the proposed task-oriented knowledge transfer approach outperforms task-agnostic transfer and CLIP pretraining approaches by a significant margin on the ImageNet dataset. See the below results.
2. Comparison with DINO Self-Supervised pretraining (as an alternative to CLIP/ImageNet pre-training) - Specifically, we pretrain the small target model on unlabeled target task data using the DINO SSL approach for 300 epochs in the case of ImageNet and Places365, 20K epochs in the case of HAM10K, and 2K epochs in the case of curated ADE20K. The proposed approach outperforms DINO SSL pretraining by a significant margin on all datasets.


OpenCLIP VFM - FastViT - ImageNet transfer

| Percentage of labeled images | 1 | 10 | 25 | 50 |
|---|---|---|---|---|
| Task-oriented transfer| 74.0 | 80.1 | 80.9 | 81.4 |
| Task-agnostic transfer | 67.7 | 74.8 | 76.0 | 78.0 |
| DINO-SSL | 52.3 | 70.6 | 74.8 | 77.4 |
| CLIP pretraining | 51.9 | 65.0 | 71.9 | 76.1 |
| From scratch | 14.5 | 56.4 | 68.9 | 75.2 |


OpenCLIP VFM - FastViT - Places365 transfer

| Labeled images per class | 50 | 250 | 1000 |
|---|---|---|---|
| Task-oriented transfer | 50.47 | 54.82 | 56.80 |
| Task-agnostic transfer | 48.83 | 51.92 | 54.74 |
| DINO-SSL | 43.44 | 50.31 | 53.79 |
| ImageNet-Pretrain | 40.58 | 47.96 | 52.33
| CLIP-Pretrain | 45.17 | 47.83 | 52.37 |


DINOV2 VFM - FastViT - HAM10K transfer

| Labeled images per class | 100 | 250 | 500 | 1000 |
|---|---|---|---|---|
| Task-oriented transfer | 78.04 | 82.30 | 86.33 | 86.2 |
| Task-agnostic transfer | 75.29 | 78.77 | 83.96 | 85.19 |
| DINO-SSL | 69.66 | 73.54 | 79.56 | 81.5 |
| ImageNet-Pretrain | 69.11 | 74.85 | 78.73 | 82.23 |
| CLIP-Pretrain | 66.64 | 72.33 | 78.31 | 80.27 |

DINOV2 VFM - FastVIT - Curated ADE20K transfer

| Labeled images | 1200 | 2401 | 4802 | 9605 |
|---|---|---|---|---|
| Task-oriented transfer | 37.65 | 40.4 | 43.28 | 44.93 |
| Task-agnostic transfer | 30.94 | 33.53 | 38.4 | 41.4 |
| DINO-SSL | 20.02 | 24.63 | 30.5 | 34.84 |
| ImageNet-Pretrain | 22.58 | 27.11 | 33.12 | 37.54 |
| CLIP Pretrain | 15.34 | 20.88 | 27.57 | 32.63 |

---

### Meta-Review · Area_Chair_r4PW · 2023-12-06

**Metareview:**

This paper proposes a label-efficient fine-tuning approach of Vision Foundation Models (VFMs) using small labeled data. To this end, the authors proposed a task-oriented fine-tuning approach, where the VFM is first fine-tuned on small labeled data, and distilled to a small model using task-agnostic unlabeled datasets. The distilled model is further fine-tuned on the small dataset to be deployed on devices with a low computation budget.

The paper initially received mixed reviews of one reject, one borderline reject, and two borderline accepts. The primary concerns raised by the reviewers are (1) justifications on problem setting (i.e., low compute budgets on inference but no such constraints on fine-tuning, which makes the other alternatives appealing such as employing an ensemble of pre-trained models during fine-tuning), (2) similarity to the existing works (i.e., the idea of distillation using the unlabeled data has been explored), (3) limited experiment setting (i.e., no ablation study on impact of domains gaps between unlabeled and target datasets), and (4) limited technical novelty.

After carefully reading the paper, reviews, and authors’ responses, AC agrees with the reviewers’ concerns. Regarding contribution, the paper clearly provides some novel insights on this specific problem setting, but they are not ground-breaking since ideas of employing the distillation for label-efficient training, cross-domain distillation, and label-efficient fine-tuning of VFMs are all well investigated in the prior works while the proposed setting is a specific combination of these problems. Regarding practicality, the proposed method assumes that the compute to fine-tune VFM is available during training, but not for inference. Although this can be reasonable in some cases, it clearly limits the scope of the work compared to existing works that employ frozen VFMs for distillation. Hence, to justify the necessity of fine-tuning, the authors should more thoroughly investigate the other appealing alternatives based on frozen VFM, such as (1) employing the frozen VFM while jointly training the student model with two objectives of distillation (on feature-level) and target task (with head) using unlabeled and labeled data, respectively, (2) employing the frozen VFM backbone features but only fine-tuning the head of VFM with target task and apply the above joint training for student model but using output head features in both distillation and target task, (3) employing an ensemble of frozen VFMs similar to (Borup et al., 2023) for distillation, etc. Considering that the core contribution of the paper is about proposing the specific problem setting and providing empirical observations on this setting, the paper needs revision to be convincing. Hence, AC recommends rejection this time.

**Justification For Why Not Higher Score:**

The justification of the problem setting is insufficient to convince the readers.

**Justification For Why Not Lower Score:**

N/A

---

### Decision · Program_Chairs · 2024-01-16

Reject